

# Integrative analysis of miRNA–mRNA network in idiopathic membranous nephropathy by bioinformatics analysis

Wenfang He[1,2,*]    Jinshi Zhang[2,*]    Shizhu Yuan[2]    Mingzhu Liang[2]    Weidong Chen[1]    Juan Jin[3]

[1] Department of Nephrology, The First Affiliated Hospital of Bengbu Medical College, Anhui, China

[2] Department of Nephrology, Zhejiang Provincial People's Hospital, Hangzhou Medical College, Hangzhou, Zhejiang, China

[3] Department of Nephrology, The First People's Hospital of Hangzhou Lin'an District, Affiliated Lin'an People's Hospital, Hangzhou Medical College, Hangzhou, Zhejiang, China

[*] These authors contributed equally to this work.

## ABSTRACT

**Background**. Currently, several specific antigens, M-type receptor for secretory phospholipase A2(PLA2R1), thrombospondin type-1 domain-containing 7A(THSD7A), and neural epidermal growth factor-like 1 protein (NELL-1), are discovered associated with the onset of idiopathic membranous nephropathy (IMN). But the pathomechanisms of IMN still need to be further claried. Understanding the mechanisms of IMN is required to improve its diagnosis and treatment.

**Methods**. In this study, we constructed miRNA regulatory networks to investigate IMN development. Moreover, miRNAs and mRNAs that were differentially expressed between Idiopathic Membranous Nephropathy (IMN) patients and normal controls were examined using the GSE115857 dataset and our previous sequence study. DE miRNA target genes were determined based on the FUNRICH software, starBase, miRDB, and miRWalk, and an miRNA-mRNA network was designed using DE-mRNAs that were negatively correlated with DE-miRNAs. The miRNA-mRNA network contained 228 miRNA-mRNA pairs. Thereafter, we conducted KEGG pathway, GO functional annotation, immune-related gene screening, protein interaction networks, and potential hub gene analyses. Furthermore, 10 miRNAs and 10 genes were determined and preliminarily validated using the validation dataset from GEO. Finally, we identified which pair may offer more accurate diagnosis and therapeutic targets for IMN.

**Results**. Two miRNA-mRNA pairs, miR-155-5p-FOS and miR-146a-5p-BTG2, were differentially expressed in IMN, indicating that these genes may affect IMN through immune processes. These findings may offer more accurate diagnoses and therapeutic targets for IMN.

Corresponding authors
Weidong Chen, cwd2012@163.com
Juan Jin, lang_018@163.com

## INTRODUCTION

Studies have shown that idiopathic membranous nephropathy (IMN) is a prevalent type of nephrotic syndrome among adults (*Ponticelli & Glassock, 2014*) and is associated with

increased prevalence, particularly in China (*Xu et al., 2016*). It is the second leading cause of end-stage renal disease (ESRD) in patients with primary glomerulonephritis (*Ronco & Debiec, 2015*). IMN is characterised by subepithelial deposits and diffusely thickened glomerular basement membranes, with clinical presentation of severe proteinuria, oedema, hypoalbuminemia, and hyperlipidaemia (*Hull & Goldsmith, 2008*). Membranous nephropathy (MN) is an organ-specific autoimmune disease. Podocytes are regarded as the primary targets of the immunological response, which often involves podocyte antigens (*Ronco & Debiec, 2020*). A few auto-antigens have been identified in adult IMN including PLA2R1, THSD7A, and NELL-1 (*Liu et al., 2020*). The understanding of the pathogenesis and diagnoses, as well as treatment, has improved because of the identification of these new indicators.

Podocyte maintenance may play an important role in the prevention and treatment of membranous nephropathy and may indirectly improve its damage by blocking the input of immune stress (*Iranzad et al., 2021*). Reducing the immune response to prevent the development of IMN by regulating immune-related genes may serve as an important breakthrough. However, several factors in IMN still require further study and clarification. Besides, given our limited understanding of the detailed mechanism of immune-associated genesin IMN development, treatment with costly and potentially toxic drugs is challenging. Therefore, exploration of IMN mechanisms is required to improve its diagnosis and treatment.

MicroRNAs are non-coding RNAs that are 21–24 nucleotides in length. MicroRNAs drive disease pathogenesis by regulating expression levels of their specific target genes (*Esteller, 2011*). Published studies have shown that miRNA dysregulation contributes to IMN development, and is therefore a potential drug target for IMN treatment (*Barbagallo et al., 2019*; *Li et al., 2019*; *Sun et al., 2020*). In our previous study, the expression patterns of urinary exosomal miRNAs varied between IMN samples and healthy control samples (*Zhang et al., 2020*). Thus, exploring the relationships between miRNAs and target genes in IMN may assist in early diagnosis and treatment.

In this study, we aimed to integrate the analysis of IMN-related miRNAs and mRNAs, acquired through the GSE115857 dataset from GEO and our previous sequence study (*Zhang et al., 2020*). A miRNA-mRNA regulatory network was established from DE-mRNAs that were negatively correlated with DE-miRNAs. Thereafter, KEGG pathway analysis, GO functional annotation, immune-related gene screening, PPI network analysis, and potential hub gene analyses were also conducted. Finally, potential hub genes and miRNAs were selected and preliminarily validated using the GSE133288 and GSE64306 datasets from GEO. The present study may help elucidate the pathogenic mechanism to improve the diagnosis, prognosis, and therapy of IMN.

## MATERIALS & METHODS

### Data source

The miRNA expression profile obtained from our previous study, which included urinary exosomal miRNAs, contained five healthy controls and six subjects with IMN (*Zhang*
_et al., 2020_). mRNA expression profiles were downloaded from the GEO database. The main inclusion criteria were as follows: (1) the dataset included kidney tissue from IMN patients and healthy volunteers and (2) the samples in each group did not undergo any specific treatment. After rational screening, GSE115857 was included in the present study. After rational screening, GSE115857, GSE133288, and GSE64306 were included in our study. The microarray data from GSE115857 were based on the GPL14951 (Illumina HumanHT-12) platform and contained seven healthy controls and 11 patients with IMN. GSE133288 comprised mRNA data and included 48 patients with IMN and five healthy controls, involving the probe platform GPL19983 (Affymetrix Human Gene 2.1 ST Array). GSE64306 comprised miRNA data and included three patients with IMN and six healthy controls whose probe platform was GPL19117 (Affymetrix Multispecies miRNA-4 Array).

## Acquisition of differential expression miRNAs and mRNAs

Differentially expressed mRNAs (DE-mRNAs) between IMN and non-IMN volunteer samples of GSE115857 were identified using the limma package in R (version 3.36.3) (_Gentleman et al., 2004_) with thresholds of |logFC|>1 and adjusted _P_-values <0.05. Differentially expressed miRNAs (DE-miRNAs) between the IMN and non-IMN volunteer samples were assessed using the EdgeR package (edgeR 3.14.0) (_Robinson, McCarthy & Smyth, 2010_) in R. Significant DE-miRNAs were selected using |logFC|>1 and adjusted _P_-values <0.05.

## The prediction of target genes of DE-miRNA and establishment of a DE miRNA-DE mRNA regulatory network

Target mRNAs of DE-miRNAs were predicted using three prediction tools: FUNRICH software (Version 3.1.3), miRWalk (http://mirwalk.umm.uni-heidelberg.de), and miRDB (http://mirdb.org/). The overlapping genes between targets and DE-mRNAs were selected, and the DE-mRNAs that negatively correlated with DE-miRNAs were further screened. The miRNA-mRNA regulatory network was created using the Cytoscape software (_Smoot et al., 2011_).

## Determination of pathways and biological processes linked to DE- mRNAs

The KEGG pathway and GO enrichment assessments were performed on DE-mRNAs in the regulatory network using Enrichr (https://maayanlab.cloud/Enrichr) (_Kuleshov et al., 2016_). The cut-off value was set at _P_ < 0.05.

## Immune-related DE- mRNAs

Immune-related genes were identified based on the InnateDB database. InnateDB is a manually curated knowledge base of genes involved in human innate immunity (_Breuer et al., 2013_). Thus, we screened DE-mRNAs in the regulatory network against immune-associated genes in the InnateDB database. Overlapped genes were considered MN-related immune DE-mRNAs and were used in the following analysis.

## PPI Networks of MN-related immune DE-mRNAs

The PPI network of DE-mRNAs was created using the STRING database (http://string-db.org/) (*Szklarczyk et al., 2011*), a tool for analysing functional interactions between DE-mRNAs. Hub genes were identified by calculating the CytoHubba plugin from the Cytoscape software (*Smoot et al., 2011*). Hub gene scores of connectivity were calculated using the Maximal Clique Centrality algorithm (MCC) which is described as:

$$MCC(v) = \sum_{C \in S(v)} (|C| - 1)!,$$

where *S(v)* is the collection of maximal cliques which contain *v*, and (|*C*|-1)! is the product of positive integers less than |C| (*Chin et al., 2014*).

# RESULTS

## Identification of DE mRNAs and miRNAs

According to an analysis of the GSE115857 dataset, we obtained 815 DE-mRNAs using the limma package in R 3.5.3 (adjusted *P*-value <0.05, | logFC|>1) (Figs. 1A, 1B). According to the analysis of our previous study (*Zhang et al., 2020*), a total of 849 miRNAs were differentially expressed between the IMN and CON groups, 43 miRNAs were significantly upregulated, and 15 miRNAs were significantly downregulated according to the criteria of adjusted *P*-value <0.05, | logFC|>1 (Figs. 1C, 1D).

## Establishment of a miRNA-mRNA network

The DE miRNA target genes were predicted using the prediction tools described above, including the FUNRICH software, miRDB, and miRWalk. The predicted target genes overlapped with DE mRNAs, while DE mRNAs that were expressed in opposition to the corresponding DE miRNA targets were selected as key DE mRNAs (Fig. S1). Finally, we obtained 139 key DE mRNAs, and designed a miRNA-mRNA regulatory network (Fig. 2A), which included 228 miRNA-mRNA pairs. The cytoHubba plugin selected the top 10 nodes with the highest score of connectivity, and 10 miRNAs were defined as hub miRNAs (Fig. 2B).

## Functional annotation analysis of key genes

The GO enrichment for 139 key genes in the regulatory network showed that the most enriched terms were "negative regulation of cellular process," "mitotic nuclear envelope reassembly," and "regulation of transcription from RNA polymerase II promoter" in the BP category; "catenin complex," "specific granule membrane," and "ionotropic glutamate receptor complex" in the CC category; and "core promoter proximal region sequence-specific DNA binding" and "RNA polymerase II core promoter proximal region sequence-specific DNA binding" in the MF category (Fig. S2). Furthermore, findings from KEGG analyses suggested that the genes were predominantly linked to the MAPK signalling pathway, osteoclast differentiation, B cell receptor signalling pathway, viral carcinogenesis, and IMN-associated pathways (Fig. S3).

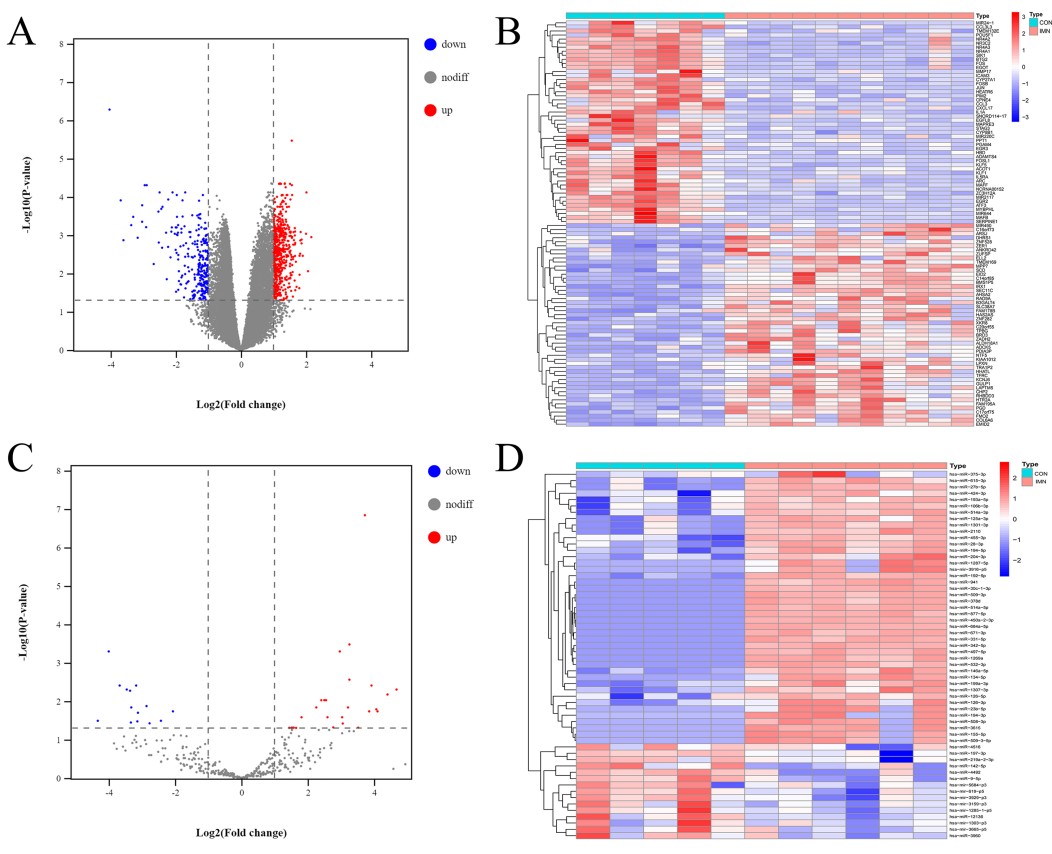

**Figure 1** **Profiling of DE mRNAs and DE miRNAs in IMN.** (A) Heat map and (B) volcano plot of DE mRNAs among IMN samples and healthy controls. (C) Heat map and (D) volcano plot of DE miRNAs among IMN patients and healthy controls. Red represents upregulation and blue represents downregulation.

## Screened immune-related genes

In further analyses, we explored the innate immune genes that were altered in IMN using the InnateDB database. A total of 59 IMN-related immune-associated genes were included in further analysis (Fig. 3A).

## PPI network of the IMN-related immune-associated genes

Next, IMN-related immune-associated genes in the PPI networks were explored to determine the hub genes. The PPI network comprised 47 edges and 59 nodes (Fig. 3B). The CytoHubba plugin in Cytoscape was used to identify hub genes, and tencandidates were selected (Fig. 3C).

## Identification of hub gene and miRNA expression using validation dataset

We demonstrated the hub miRNA using the GSE64306 dataset. We focused on the expression levels of hsa-miR-155-5p, hsa-miR-4516, hsa-miR-1269a, hsa-miR-4492, hsa-miR-197-3p, hsa-miR-9-5p, hsa-miR-146a-5p, hsa-miR-23b-5p, hsa-miR-194-3p,
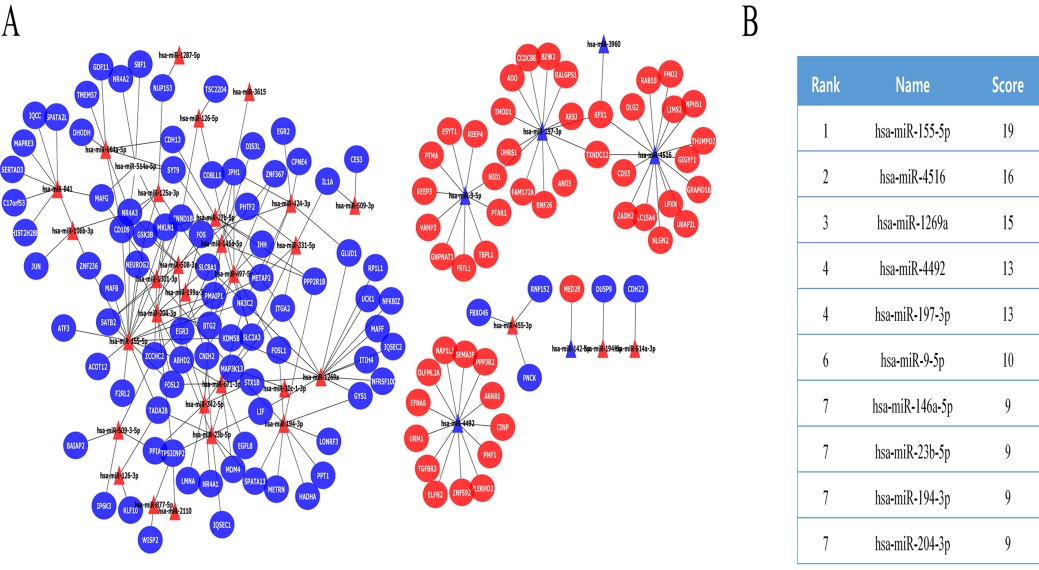

**Figure 2  The miRNA-mRNA regulatory network in IMN and interaction between 10 hub miRNAs and target DE mRNAs.** (A) The miRNA-mRNA regulatory network in IMN. The triangles represent DE-miRNAs, and the ellipses represent target DE-mRNAs. Red indicates upregulation and blue indicates downregulation. (B) The interaction between 10 hub miRNAs and target DE mRNAs.

and hsa-miR-204-3p. As a result, we found that hsa-miR-155-5p, hsa-miR-146a-5p, and hsa-miR-4492 showed significant expression changes in the validation dataset (Fig. 4). Another mRNA expression profile dataset GSE133288 from GEO was used for hub mRNA validation. We investigated the expression levels of *FOS, JUN, FOSL1, BTG2, NR4A1, MAFB, MAFG, HIST2H2BE,* and *IL1A.* We observed that *FOS, JUN, BTG2,* and *NR4A1* showed significant differential expression in the validation dataset (Fig. 5). Comparing the two results, we observed that two miRNA-mRNA pairs, miR-155-5p-*FOS* and miR-146a-5p-*BTG2*, were differentially expressed in IMN.

## DISCUSSION

Numerous studies have shown that immune and epigenetic processes influence the development of IMN (*Roccatello et al., 2016*; *Sha et al., 2015*). Among epigenetic disruptions, changes in miRNA levels may alter immune functions and hence affect the prognosis of IMN. Understanding the impact of abnormal miRNA expression in IMN will help in developing strategies for diagnosing and treating IMN (*Barbagallo et al., 2019*; *Motavalli et al., 2021*).

Previous studies have shown that urinary exosomes contain miRNAs originating from the tubular and glomerular cells (*Salih, Zietse & Hoorn, 2014*). Here, we comprehensively assessed the differential expression levels of miRNA in urinary exosomes and mRNA in the kidney tissues of IMN patients, 43 DE miRNAs and 815 DE mRNAs. We further predicted the DE miRNA target genes and overlapped them with DE mRNAs, and then constructed the miRNA-mRNA regulatory network, which was composed of 139 key genes and 228

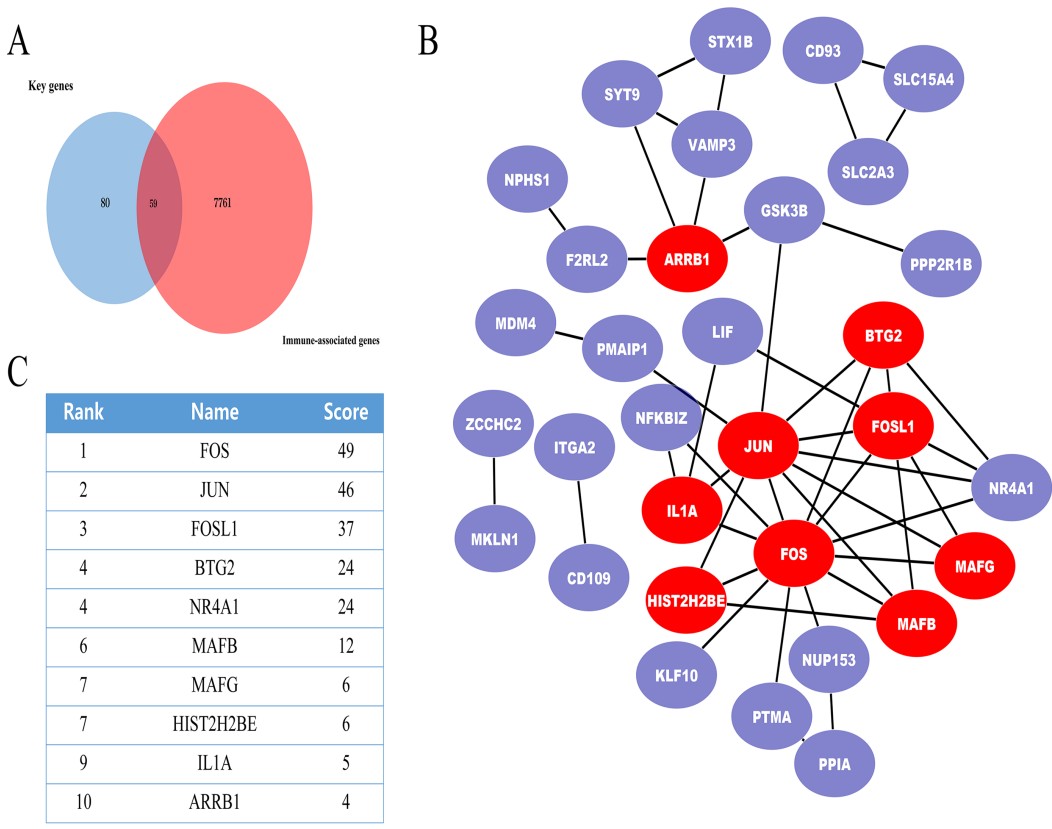

| Rank | Name | Score |
|------|------|-------|
| 1 | FOS | 49 |
| 2 | JUN | 46 |
| 3 | FOSL1 | 37 |
| 4 | BTG2 | 24 |
| 4 | NR4A1 | 24 |
| 6 | MAFB | 12 |
| 7 | MAFG | 6 |
| 7 | HIST2H2BE | 6 |
| 9 | IL1A | 5 |
| 10 | ARRB1 | 4 |

**Figure 3** **The miRNA-mRNA regulatory network in IMN and top ten miRNAs with the highest score of interaction in the miRNA-mRNA regulatory network.** (A) The miRNA-mRNA regulatory network in IMN. The triangles represent DE-miRNAs, and the ellipses represent target DE-mRNAs. Red indicates upregulation and blue indicates downregulation. (B) The interaction between 10 hub miRNAs and target DE mRNAs. (C) Top ten miRNAs with the highest score of interaction in the miRNA-mRNA regulatory network.

miRNA-mRNA pairs. For the 139 key genes in the regulatory network, three GO_BP terms (such as negative regulation of cellular process), three GO_CC terms (such as catenin complex), and three GO_MF terms (such as RNA polymerase II core promoter proximal region sequence-specific DNA binding) were enriched.

Results from KEGG pathway analyses demonstrated that several important genes were enriched in the MAPK, B cell receptor, and osteoclast differentiation pathways. Among these pathways, the B cell receptor pathway is known to influence the B cell signalling network and stimulate autoimmune disease pathogenesis, resulting in IMN (*Motavalli et al., 2019*). The MAPK signalling pathway also plays a significant role in the immunomodulatory effects of T cells and B cells (*Dennison, Mohan & Yarchoan, 2021*). The results of functional annotation analysis suggest that immune-related genes warrant future investigation.

Therefore, we screened 139 key genes that correlated with immunity using the InnateDB database. A total of 59 immune-associated genes were screened to construct a PPI network. The CytoHubba plugin of Cytoscape was used to analyse the miRNA-mRNA regulatory

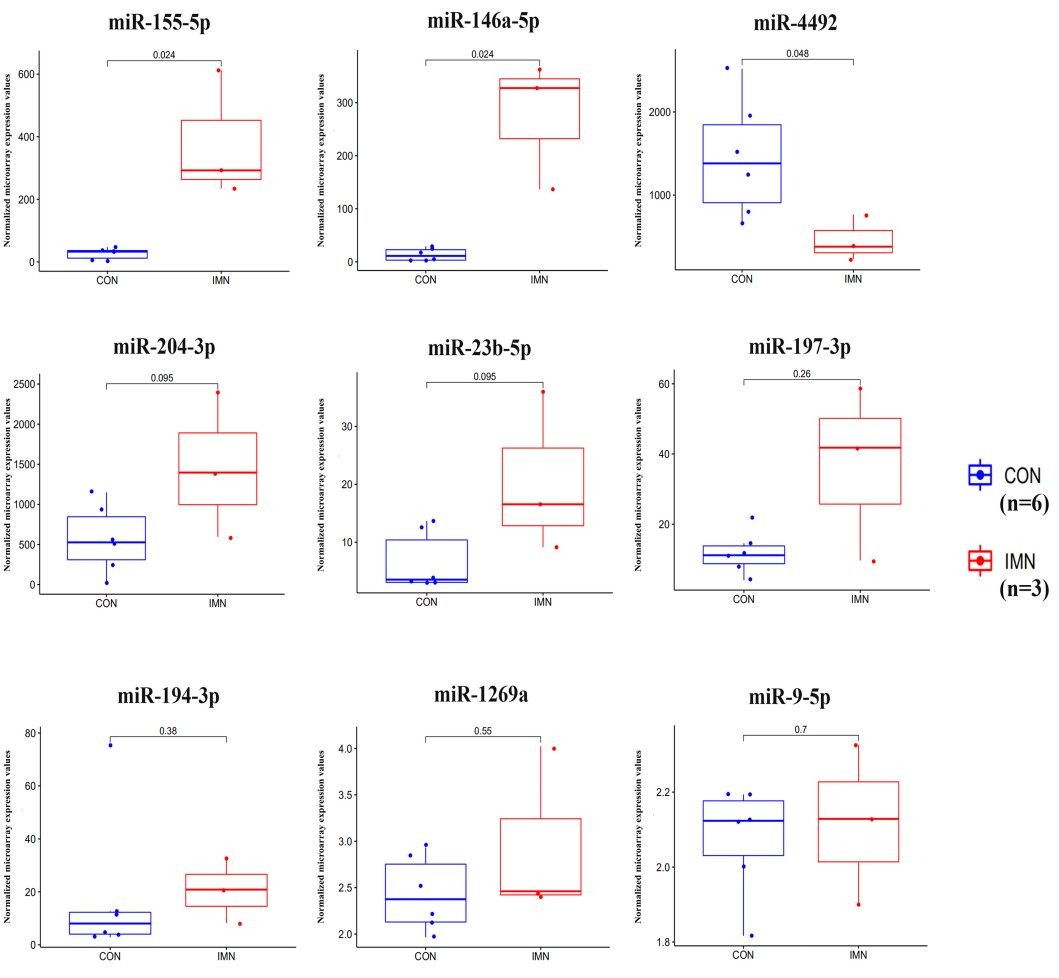

**Figure 4** **Expression profile of hub miRNAs in the validation database (GSE64306).**

network and the immune-associated gene PPI network. Tenhub genes and 10 hub miRNAs were screened. Next, the hub miRNAs and genes were identified by analysing these against the validation dataset. Finally, two miRNA-mRNA pairs, miR-155-5p-*FOS* and miR-146a-5p-*BTG2*, were found to affect immune processes and offer more accurate diagnosis and therapeutic targets for IMN.

MiR-155-5p is a multifunctional miRNA which is overexpressed in the kidney tubules of diabetic kidney and renal fibrosis patients (*Baker et al., 2017*). Recent research has revealed that miR-155-5p can cause oxidative stress and inflammation in diabetic kidneys and renal fibrosis by attenuating the autophagic process (*Guo et al., 2020*; *Wang et al., 2020*; *Wang et al., 2018*). Our previous study showed that autophagy inhibition aids IMN-induced podocyte injury (*Jin et al., 2018*). In summary, miR-155-5p may be essential for the regulation of the autophagic process in podocytes of IMN.

Genes in *FOS* encode leucine zipper proteins, which are involved in the dimerisation of proteins belonging to the JUN family. Thus, they lead to the formation of transcription factor activator protein (AP-1). *FOS* proteins modulate differentiation, proliferation,

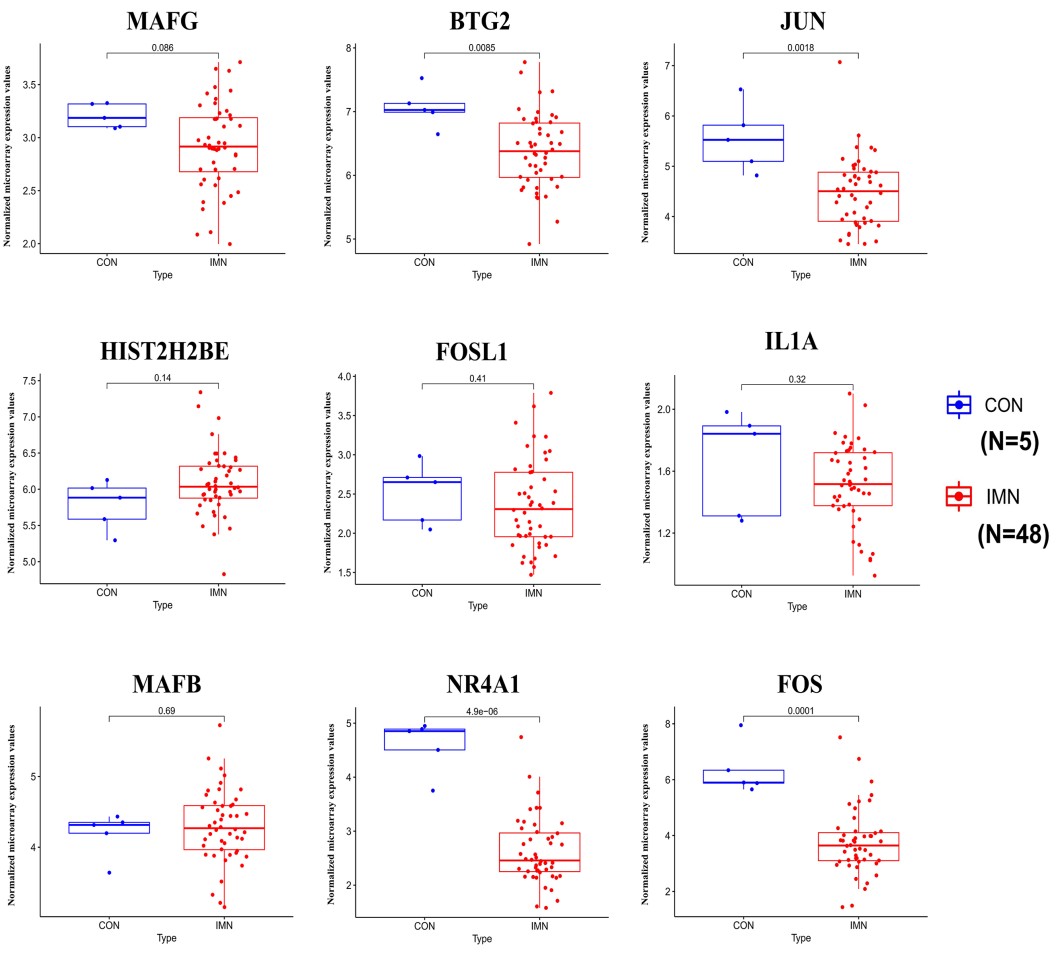

**Figure 5** **Expression profile of hub genes in the validation database (GSE133288).**

and apoptotic cell death (*Tang et al., 2020*). Previous investigations have shown that the *FOS* gene is also involved in immune system regulation (*Wagner & Eferl, 2005*). Recent research has revealed that *FOS* can affect the transcription of inflammatory cytokines by binding to their promoters, leading to their high expression inducing endotoxin-induced and cisplatin-induced acute kidney injury (AKI) (*Miyazaki et al., 2012*; *Zhang et al., 2019*). There is considerable evidence that inflammatory processes contribute to the pathogenesis of IMN (*Nangaku, Shankland & Couser, 2005*; *Wu et al., 2008*). These results confirmed the association between *FOS* and IMN.

MiRNA-146a-5p was first reported in humans in 2006 (*Taganov et al., 2006*) and is a well-known miRNA associated with inflammation (*Rusca & Monticelli, 2011*). It was revealed that miR-146a expression is upregulated in the kidney, where it induces chronic inflammation (*Ichii et al., 2012*). The effects of miR-146a-5p have been demonstrated in renal diseases and can be used for the treatment of renal fibrosis and ischemia/reperfusion injury by inhibiting the infiltration of inflammatory cells (*Li et al., 2020*; *Morishita et al.,*

*2015*). From this, we can speculate that miRNA-146a-5p may influence IMN through its effects on inflammation.

BTG2 belongs to the B cell translocation gene in the BTG/Tob family, which is associated with several cellular processes (*Mauxion et al., 2009*; *Winkler, 2010*; *Yuniati et al., 2019*). It is noteworthy that BTG2 was highly expressed in lymph nodes and white blood cells, suggesting its role in the immune system. BTG1 deficiency leads to inappropriate T-cell clonal expansion, while differentiation can cause immune-related diseases (*Hwang et al., 2020*). Consistent with the literature, our research found that the expression of BTG2 was abnormally low in patients with IMN.

There are some limitations in our study. The datasets GSE115857, GSE133288, and GSE64306 were obtained from the GEO database and were generated by different researchers, this may result in certain biases during gene analysis. Further, as shown in some similar studies, we only showed the opposites interacting of miRNAs and mRNA. miRNA-mRNA regulatory network in IMN were based on target gene prediction and statistical evidence, further experimental validation is required to verify the results.

## CONCLUSIONS

We present a miRNA-mRNA regulatory network consisting of 228 miRNA-mRNA pairs in IMN. We found that two miRNA-mRNA pairs, miR-155-5p-*FOS* and miR-146a-5p-*BTG2*, were differentially expressed in IMN, indicating that these genes may affect IMN through immune processes. These findings may offer more accurate diagnoses and therapeutic targets for IMN. Further experimental validation is required to verify the results.

## ACKNOWLEDGEMENTS

We would like to thank Editage for English language editing.

### Funding

This research was supported by the Construction of Key Projects by Zhejiang Provincial Ministry (Grant Number: WKJ-ZJ-1915), the General Project of Zhejiang Education Department (Grant Number: Y201942823), and the Clinical and Experimental Research of YSHS Granule. The funders had no role in study design, data collection and analysis, decision to publish, or preparation of the manuscript.

### Grant Disclosures

The following grant information was disclosed by the authors:
Zhejiang Provincial Ministry: WKJ-ZJ-1915.
General Project of Zhejiang Education Department: Y201942823.
Clinical and Experimental Research of YSHS Granule.

### Competing Interests

The authors declare there are no competing interests.

 

## Author Contributions

- Wenfang He conceived and designed the experiments, performed the experiments, authored or reviewed drafts of the paper, and approved the final draft.
- Jinshi Zhang conceived and designed the experiments, performed the experiments, prepared figures and/or tables, authored or reviewed drafts of the paper, and approved the final draft.
- Shizhu Yuan and Mingzhu Liang analyzed the data, prepared figures and/or tables, and approved the final draft.
- Weidong Chen and Juan Jin conceived and designed the experiments, authored or reviewed drafts of the paper, and approved the final draft.

## Data Availability

The expression data is available at GEO: GSE115857, GSE133288, GSE64306.

## Supplemental Information

Supplemental information for this article can be found online at http://dx.doi.org/10.7717/peerj.12271#supplemental-information.

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
