# Peer review of "Integrative analysis of miRNA–mRNA network in idiopathic membranous nephropathy by bioinformatics analysis"

_PeerJ, doi:10.7717/peerj.12271_

## Round 0.1 · original submission · Major Revisions

Please modify figures as suggested by reviewers (and take into account to reduce their number).

Reviewer 1 ·

Basic reporting

The author's goal is to integrate publicly-available mRNA and miRNA datasets to identify potential biomarkers or therapeutic targets for idiopathic membranous nephropathy. Once they have identified and analyzed 2 datasets, they seek to confirm their findings in validation datasets. The basic flow of the manuscript is to identify differentially expressed (DE) mRNAs and miRNAs, find the DE mRNAs that are negatively correlated with the DE miRNAs, conduct biological annotation and pathway analysis and then validate their findings.

The authors should be applauded for their ability to take published data, re-analyze it and make novel observations. This approach should be more frequently used by other investigators as the amount of data available currently greatly exceeds the amount of in-depth analysis available. In addition, integrating 'omic data is currently a hot research area. However, just because an analysis can be done doesn't mean it should be done or that it contributes to the goals of the manuscript. These are my major areas of concern with this manuscript, namely, that it is short on biological or clinical context and it isn't always clear why (and sometimes how) an analysis was performed. More details in the comments to the authors.

In general, the manuscript is well-written, but there are areas that are not idiomatic. For example, the authors say that genes are 'remarkably' expressed when the general term used is 'significantly'.

Experimental design

As stated, I like the approach of re-analyzing public data and integrating miRNA and mRNA data to discover novel biological insights. However, there are a number of areas where greater detail and/or concern exist:

Validity of the findings

Please see below.

Additional comments

Figure 1- The labels for volcano plots and heatmaps are incorrect.
Why does 1D use a different color scheme?
For figure 1 the authors state that 3415 miRNAs were differentially expressed, while 43 were 'remarkably' up-regulated and 15 were down-regulated. I don't know where the 3415 number comes from or what it means, but I think that mammals are only believed to have ~2000 miRNAs.

Figure 2- The role of the 'immune-associated genes' is one of the areas where the authors need to provide more detail and context.
One presumes that the genes in the red circle of 2A come from the analysis performed using Funrich, miRDB and miRwalk. Did the authors add the results from these three tools together? If so, why? Much, much more detail is needed here. Why did they need to use 3 tools for this analysis?
In the text, the 139 'key genes' are described as those that overlap between the predicted targets and the DE mRNAs, but also as those that are inversely correlated with the miRNAs. Both can't be true. Again, more details and context are needed here to fully understand the author's thought process.
It also seems the authors just picked DE mRNAs that were expressed in the opposite direction as the miRNAs. This is the best they can do because the datasets were not generated at the same time, but it should be described as a significant limitation.
The text says 58 genes came out of this analysis, but the figure says 59.

Figure 3- The authors describe the red and green colors, but what does yellow mean? There is no description of the colors used in 3C and a more thorough discussion of the meaning of the results is needed.
Figure 3 is a figure that I also question how it contributes to the manuscript. It is clear these networks can be generated, but it isn't clear how they are supporting the results of the study.

Figure 4- Again, colors are used with no explanation. In the text, the authors recite the results they get, but what does it mean that these pathways were found? Anythings?
Basically no context is given for the bottom part of figure 4. The one line title really lacks the information necessary to fully understand this figure.

Figure 5- much like figure 4 it needs much more explanation for the reader to understand why it is included.

Figure 6B- Why is the interaction of hub genes important?

Figure 7- In the .pdf version, figure 7 is basically unreadable while figure 8 looks really nice. The authors should remake figure 7 as they did in figure 8.

Figure 8- the y-axis needs to be labeled. The authors should double-check their statistics as well. miR-155-5p and 146a-5p are exactly the same, as are miR-204-3p and miR-23b-5p. It seems unlikely.

Figure 9- I really am not sure what this figure is trying to tell me. There needs to be a lot more detail.

Reviewer 2 ·

Basic reporting

The authors have conducted a study with paired miRNA and mRNA analyses. The language is clear and they also report having used a professional editing service. The overall design and reporting is appropriate.

Minor point: The introduction is too thin and it is not true that there is no knowledge about the pathophysiological mechanisms: Line 22-24 in abstract and line 51-52 in introduction. Please also mention and refer to the autoantigens PLA2R and THSD7.

The article has more figures than needed to tell the story. Figure 2, 4 and 5 could be shown as supplemental figures instead as they doesn't add that much anyways.

Experimental design

The experimental design seems appropriate with integration of different miRNA and mRNA datasets and appropriate analyses and figures.

Minor point: The normal control biopsies should be explained in more detail. Are these per indication biopsies that turned out to be "normal" after pathological evaluation or are they true normal biopsies of healthy volunteers?
Minor point: Some of the figures have low resolution and if possbile should be uploaded as a scalable digital format rather than picture format - otherwise the networks interpretation for the readers will be poor.

Validity of the findings

The findings are not validated with qPCR or IHC/protein level, but MiR-155-5p has been described in other renal diseases, in inflammation and in kidney cancer. Thus the findings in this study is probably correct. The authors also point to this in their conclusion. How the overexpression of MiR-155-5p is believed to implement the diagnosis of IMN is more unclear (line 240).

---

## Round 0.2 · Minor Revisions

In addition to the reviewer's new comments, please carefully correct the typo and grammar errors.

.

Reviewer 1 ·

Basic reporting

The authors have significantly improved the manuscript and largely addressed the reviewer's concerns. They do note that they used an English editing service, but it doesn't appear that this was applied to the new sections of the manuscript as they contain many errors. Overall, I have minor concerns as outlined below:

1) The supplemental figures do not have legends.

2) Figure 2A- every miRNA/mRNA pair is in opposition. This is the predicted mechanism of action, but one wouldn't expect perfectly clean data. That is, there are likely some mRNA that are down-regulated when the miRNA is down-regulated, etc. Did the authors only show the interactions that were in opposition? If so, this should be noted.

3) Figure 2B- the 'Score' is not defined. In Figure 3B the 'highest degree of interaction' is mentioned, but this should still be better defined for the reader.

4) Figure 3 doesn't mention that these are immune-related genes anywhere in the legend.

5) Related to the above concern, in the text the authors describe the 'degree' and the table talks about the 'score' of the interaction. The text also mentions that 9 hub genes had scores >5, but then a 10th gene with a score of 4 is shown. This should be removed or the text changed.

6) GSE33288 is listed and it should be GSE133288.

7) Figure 4- in the text the authors state that FOS, JUN, BTG2 and NR4A1 were significant, but IL1A and MAFB were shown to be significant in the figure as well.

8) Figure 4 and 5- the y-axis is 'relative expression'. Relative to what? I don't believe they are actually showing relative expression, but probably normalized microarray values.

9) As miRNAs are thought to regulated mRNAs, I think it makes more sense to flip figures 4 and 5 in the order.

Experimental design

See comments above

Validity of the findings

See comments above

Additional comments

See comments above

Reviewer 2 ·

Basic reporting

The reporting is adequate and the authors have corrected the introduction part regarding the background. Although, there are a few typographical errors that have been included in the new version that need to be sorted out during the proofs.

Experimental design

My concerns for the paper regarding figures have been corrected.

Validity of the findings

No concerns.

---

## Round 0.3 · accepted · Accept

The revisions made by the authors adequately answered all the reviewers' suggestions and the manuscript is, now, improved.
No further comments.